# Water Body Mapping Using Long Time Series Sentinel-1 SAR Data in Poyang Lake

**Guozhuang Shen** [1,2,*,†] , **Wenxue Fu** [1,2,†] , **Huadong Guo** [1,2,†] and **Jingjuan Liao** [1,2,†]

1   International Research Center of Big Data for Sustainable Development Goals, No. 9,
    Dengzhuang South Road, Haidian District, Beijing 100094, China; fuwx@aircas.ac.cn (W.F.);
    guohd@aircas.ac.cn (H.G.); liaojj@aircas.ac.cn (J.L.)
2   Key Laboratory of Digital Earth Science, Aerospace Information Research Institute,
    Chinese Academy of Sciences, No. 9, Dengzhuang South Road, Haidian District, Beijing 100094, China
*   Correspondence: shengz@aircas.ac.cn
†   These authors contributed equally to this work.

**Abstract:** Mapping water bodies with a high accuracy is necessary for water resource assessment, and mapping them rapidly is necessary for flood monitoring. Poyang Lake is the largest freshwater lake in China, and its wetland is one of the most important in the world. Poyang Lake is affected by floods from the Yangtze River basin every year, and the fluctuation of the water area and water level directly or indirectly affects the ecological environment of Poyang Lake. Synthetic Aperture Radar (SAR) is particularly suitable for large-scale water body mapping, as SAR allows data acquisition regardless of illumination and weather conditions. The two-satellite Sentinel-1 constellation, providing C-Band SAR data, passes over the Poyang Lake about five times a month. With its high temporal-spatial resolution, the Sentinel-1 SAR data can be used to accurately monitor the water body. After acquiring all the Sentinel-1 (1A and 1B) SAR data, to ensure the consistency of data processing, we propose the use of a Python and SeNtinel Application Platform (SNAP)-based engine (SARProcMod) to process the data and construct a Poyang Lake Sentinel-1 SAR dataset with a 10 m resolution. To extract water body information from Sentinel-1 SAR data, we propose an automatic classification engine based on a modified U-Net convolutional neural network (WaterUNet), which classifies all data using artificial sample datasets with a high validation accuracy. The results show that the maximum and minimum water areas in our study area were 2714.08 km$^2$ on 20 July 2020, and 634.44 km$^2$ on 4 January 2020. Compared to the water level data from the Poyang gauging station, the water area was highly correlated with the water level, with the correlation coefficient being up to 0.92 and the $R^2$ from quadratic polynomial fitting up to 0.88; thus, the resulting relationship results can be used to estimate the water area or water level of Poyang Lake. According to the results, we can conclude that Sentinel-1 SAR and WaterUNet are very suitable for water body monitoring as well as emergency flood mapping.

**Keywords:** Poyang Lake; Sentinel-1; SAR; U-Net; water area; water level

## 1. Introduction

As part of the Earth's hydrosphere, water resources are important resources for terrestrial life and play an important role in nature and human society. Nevertheless, surface water around the world is undergoing spatial and temporal changes caused by many factors, such as land use/cover changes, climate changes, seasonal changes, and environmental changes [1]. Meanwhile, the high-precision mapping of open water can also provide useful information for flood monitoring, assessment, and disaster relief decision-making [2], as well as for studying the relationship between the hydrological conditions and the ecological environment. Therefore, it is necessary to monitor water body changes in a timely manner and with accurate information [3].

Remote sensing technology has become indispensable in monitoring changes in water bodies due to its characteristics of high spatio-temporal coverage [4]. Even with well-defined methods and with the readily available data from optical sensors for surface water mapping applications [5], its use is hindered by clouds that obscure surface observations [6]. As an active sensor, Synthetic Aperture Radar (SAR) is particularly suitable for water body mapping, as this tool allows data acquisition regardless of illumination and weather conditions [7,8], and SAR data are also sensitive to water bodies due to the weak backscatter from the water surface [8–10].

Over the past few decades, data from SAR operating at different wavelengths and multi-mode image configurations have been used for water body mapping, including ENVISAT-ASAR [8,9,11–14], RADARSAT/RADARSAT-2 [15–22], JERS-1/ALOS-PALSAR/ALOS2 [23,24], and Chinese GF-3 [2]. Since the launch of Sentinel-1A on 3 April 2014, and Sentinel-1B on 25 April 2016, Sentinel-1 has become the main SAR data source for water body mapping, due to its 5~7-day revisit cycle and its free data policy [7,25–31]. Moreover, various water body mapping methods have been developed for SAR data. As a result of the radar's unique response to water, water body mapping using intensity thresholding methods on SAR images had been extensively used [21,22,30,32]. Some modified Otsu thresholding [33] methods that can automatically select a threshold for surface water extraction have also been proposed for extracting water body data [11,22,32,34]. Object-based image analysis (OBIA) methods have also been proposed for extracting water body information for SAR data, taking into account the shape and color information of the segmented objects; however, OBIA methods are generally computationally expensive [8,16,35,36]. Meanwhile, the time-series Sentinel-1 water index (SWI) has also been used to extract water surface areas from Sentinel-1 SAR data using the threshold method [37].

Sentinel-1 is a two-satellite constellation providing wide-swath C-Band SAR data, with data being regularly acquired every 5 to 7 days over the same place, making large-area water monitoring feasible [29]. The availability of repeated acquisitions is advantageous, since multi-temporal observations allow the detection of trends in water bodies [10,12]. More and more researchers have used time-series wide-swath Sentinel-1 SAR data to extract water surface data automatically or semi-automatically. On the premise of obtaining water and non-water mask data, fully automatic algorithms (Bayesian based thresholding [27], Support Vector Machine (SVM) [4], and Random Forest classification [5,38]) have been developed to derive water probability and classify the extent of surface water using Sentinel data [27]. For Sentinel-1 SAR data, some surface water automatic mapping methods have been proposed for use in rapid flood mapping and event maps with increasing resolution [25,26,28]. However, to monitor and analyze water body changes over a long period of time, it has become popular to remote sensing big data from Sentinel-1 SAR. Many studies have deployed algorithms on Google Earth Engine (GEE) to extract water body over long periods of time [1,31,39,40], due to its data storage advantages and data processing capabilities.

Since the idea of deep learning was proposed by Geoffrey Hinton [41] in 2006, deep learning has attracted more and more attention, and deep Convolutional Neural Network (CNNs) has also shown advantages in the field of image semantic segmentation [42,43]. As commonly used models in deep learning, CNNs have become tremendously popular in recent years because CNNs can learn extremely complicated hierarchical features from massive amounts of data [42], greatly reduce the number of parameters, enhance the generalization ability, and realize the qualitative task of image recognition [44]. CNN can learn high-level context features through a large number of neurons [45], and extract features automatically, robustly in terms of image size and context among other aspects [46], which overcomes many limitations of traditional classification methods. Fully Convolutional Network (FCN) [47], SegNet [48], RefineNet [49], and U-Net [50], which include encoding and decoding blocks to perform pixel-wise image segmentation, have become the desired models for use in image segmentation tasks; in particular, FCN and U-Net are widely used in remote sensing water body extraction [51–54], where the encoder is

used to extract image features and reduce image dimensions, and the decoder is used to restore the size and detail of the image [55]. In U-Net, the low-resolution information in the encoding blocks (down-sampling, providing the basis for object category recognition) and the high-resolution information in the decoding blocks (up-sampling, providing the basis for accurate segmentation and positioning) are combined to provide detailed information for image segmentation. In addition, skip–connection is used to fill in the low-level information to improve the segmentation accuracy in U-Net. The compact structure of the U-Net model (small size and fewer parameters) also makes it suitable for water body extraction [46].

As the largest freshwater lake in China and its wetland as one of the most important in the world, Poyang Lake is in the middle reaches of the Yangtze River and plays an important role in the ecosystem of the Yangtze River watershed. It is a typical river-connected lake, and the water body of the lake experiences a remarkable seasonal variation due to the influence of the monsoon climate of the region. However, the Three Gorges Dam operation has altered the seasonal inundation pattern of Poyang Lake and significantly reduced the monthly inundation frequencies [56,57]. The fluctuation of the water body directly or indirectly affects the ecological environment of Poyang Lake—e.g., the distribution of vegetation communities [58,59] and the spatio-temporal variation in Siberian Crane habitats [60]. Meanwhile, serious floods occasionally occur in the Poyang Lake—e.g., in 1998 and 2020. Thus, changes in the water body of Poyang Lake have always attracted the attention of researchers [29,40,61,62].

Defining an accurate and reliable method for monitoring water body changes, studying the relationship between water body surface area and water level, and studying wetland environment changes using Sentinel-1 SAR data are a challenge. Monitoring models using deep neural networks provide an ideal solution for monitoring water bodies. Here, in order to explore the potential of the use of Sentinel-1 in water body mapping and to study the relationship between the water area and water level for Poyang Lake, long-time-series Sentinel-1 SAR data were collected to extract water body information using the U-Net deep learning model. Water level data were also collected from Poyang gauging station. In this study, our work included: (1) establishing a large-scale water classification method based on the U-Net CNN model, (2) extracting Poyang Lake water body data from 2014 to 2021 and evaluating their accuracy, and (3) analyzing the inundation frequency distribution and the relationship between the water level and the water area.

## 2. Materials and Methods

### 2.1. Study Area

The study area was Poyang Lake (115°47' E~116°45' E, 28°22' N~29°15' N), Jiangxi province, China, in the lower Yangtze River Basin (Figure 1). Poyang Lake is the largest freshwater lake in China and constitutes a major hydrological subsystem of the middle Yangtze basin [29]. The climate is characterized as a sub-tropical, humid monsoon climate with 1620 mm mean annual precipitation and an annual average temperature of about 17 °C [2,63]. Poyang Lake receives inflows from five rivers—namely, the Xiushui, Ganjiang, Fuhe, Xinijiang, and Raohe [37].

Poyang Lake exhibits large inter-annual variations in its water level (Figure 2), which reflects in changes of the water surface area. The water level rises during the wet season, and the water surface expands. In the wet season, it is the largest freshwater body in China and can cover up to 3500 km$^2$ by the end of the season (April to September). In the dry season (October to March), the water level drops and Poyang Lake becomes smaller than 1000 km$^2$, with several small lakes remaining.

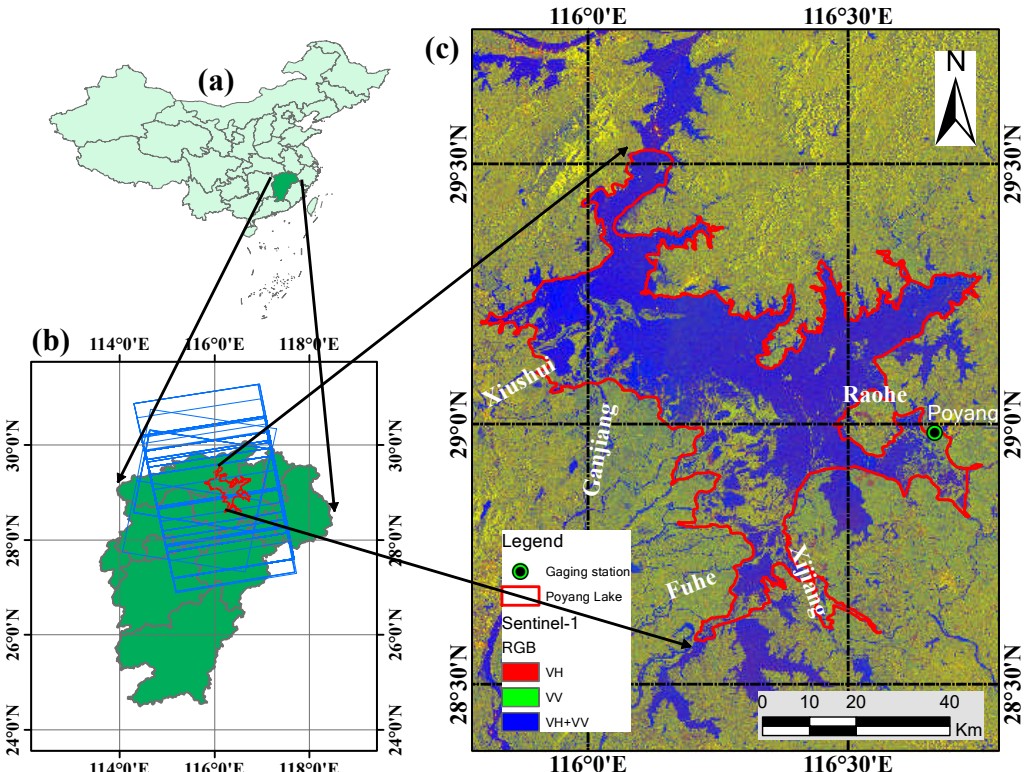

**Figure 1.** Spatial location and extent of the study area of Poyang Lake in Jiangxi Province, People's Republic of China. (**a**) shows the administrative divisions of China, the green polygon in (**b**) shows Jiangxi Province, and (**c**) shows Poyang Lake. The blue polygons in (**b**) show the swaths of Sentinel-1 SAR data and the red polygon in (**c**) shows the study area.

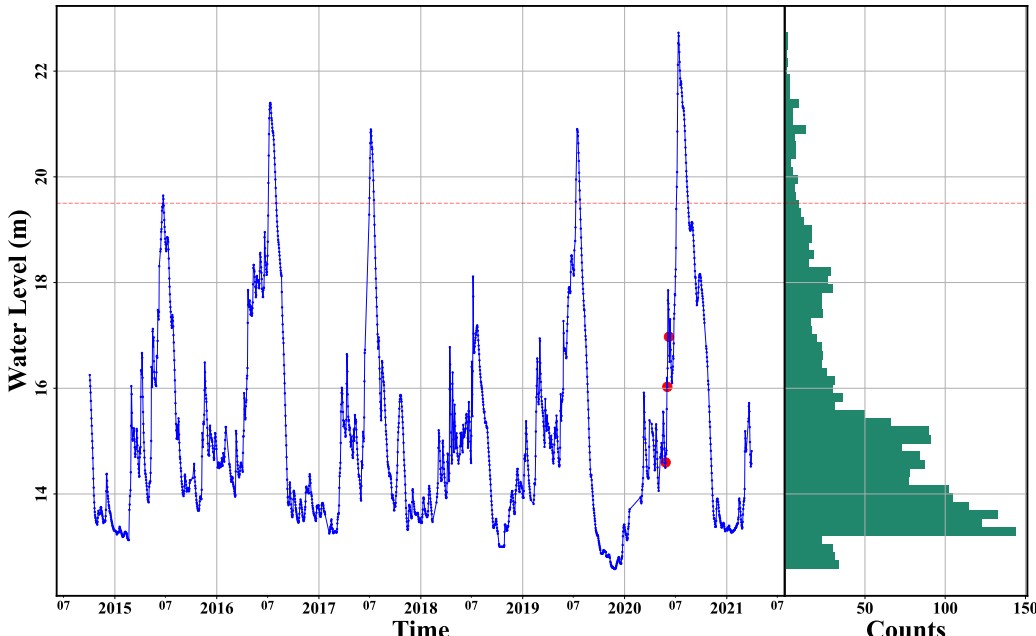

**Figure 2.** The water level data and a statistic histogram for Poyang gauging stations in Poyang Lake from October 2014 to March 2021 (Wusong water level datum). Some data are missing due to the maintenance of the website. The red horizontal line represents the warning water level, 19.5 m. The blue line is for the trend of water level, and the red dots are for water level records on 27 May 2020, 2 June 2020, and 8 June 2020. The green bars are for the count in each water level interval (64 bins).

## 2.2. Data

### 2.2.1. The Sentinel-1 SAR Data

The rainy and cloudy weather in Poyang Lake results in limited high-quality cloud–free optical images, such as from the non-commercial Landsat or Sentinel-2 platforms, to provide enough data for dynamic monitoring of changes in water surface areas of Poyang Lake [37,40]. Thus, to extract water body data for Poyang Lake and monitor the dynamics in the study area over the past 7 years, 455 scenes of the level-1 Ground Range Detected (GRD) Sentinel-1 C-band (5.405 GHz) SAR data from the Interferometric Wide swath (IW) and VV/VH polarization mode were collected from October 2014, to March 2021 from Copernicus Open Access Hub (https://scihub.copernicus.eu/, accessed on 18 April 2021). The blue polygons in Figure 1b show the swaths for these 455 scenes of SAR data. After Sentinel-1 slice assembly, 455 scenes of Sentinel-1 SAR data formed 324 effective observations due to the slice assembly of the two consecutive 'slices'. In these 324 SAR observations, there were 187 observations from Sentinel-1A and 137 from Sentinel-1B, with nine observations operating in descending mode and 315 observations in ascending mode (Figure 3).

The temporal distribution of the acquisition dates and the numbers for these 324 SAR observations are shown in Table A1 and Figure 4. From Table A1, we can see that, as the launch of Sentinel-1B on 25 April 2016, the observation times in one year increased and could reach up to 60 times (in the years 2017, 2018, 2019, and 2020). However, due to the commissioning phase of Sentinel-1A, there are not available data from January 2015 to March 2015. From Figure 4, we can also see that (1) the minimum number of observation times for all 75 months is 1 (April, May, and November 2015; June 2016), (2) the maximum number of observation times is six (January 2017, May 2017, July 2018, August 2019, and August 2020), and (3) there are five observations in most months.

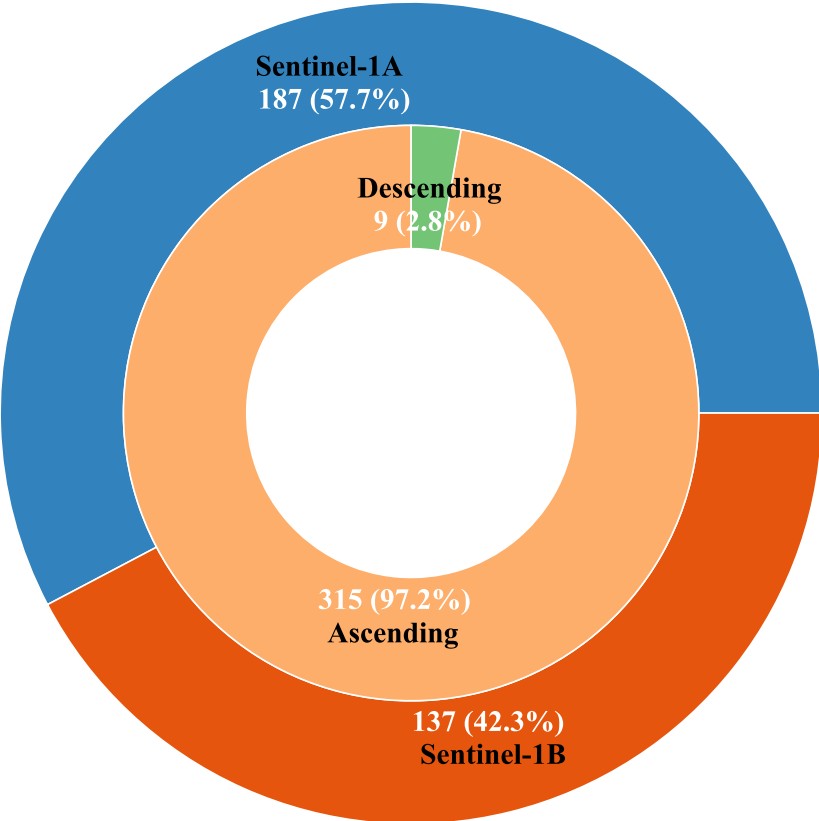

**Figure 3.** The number and percentage of the observations for different satellites and overpasses.

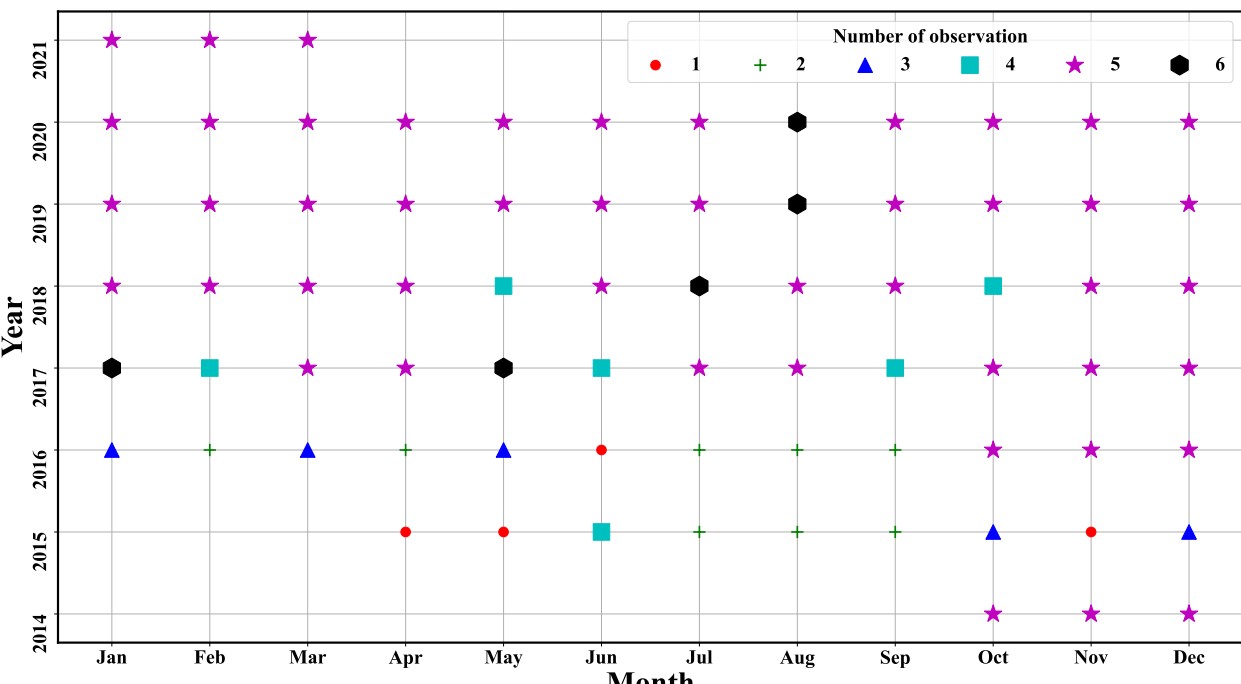

**Figure 4.** The temporal distribution and number of Sentinel-1 SAR observations per month between 2014 and 2021.

#### 2.2.2. Water Level

In this study, the daily water level data (Figure 2) monitored at Poyang gauging station (Figure 1c) were collected from the website of the Hydrological Bureau of Jiangxi Province (http://www.jxssw.gov.cn/, accessed on 1 July 2021). The time period selected was from 2014 to 2021, which coincides with the time range of the Sentinel-1 SAR data. The water level was automatically measured using pressure transducers or noncontact transducers based on the reference of elevation systems of Wusong water-level datum [64]. Figure 2 shows the interannual and seasonal fluctuation of the water level from Poyang gauging station. The lake's water level fluctuated regularly from 2014 to 2021, reaching the annual highest water level in the wet season in July every year. There is a secondary water level peak in March or April each year before the highest water level. Furthermore, the highest water level occurred on 12 July 2020, at 22.73 m, when Poyang Lake was suffering from the most severe flood in the study period. The warning water level for Poyang Station is 19.5 m, and the water level exceeded the warning level five times, in 2015, 2016, 2017, 2019, and 2020. The lowest water level occurred on 27 November 2019, at 12.58 m. From Figure 2, we can also see that 50% of the water level values are lower than 15 m.

#### 2.3. Methodology

The flowchart of this study is shown in Figure 5. This flowchart contains four steps: SAR image pre-processing (SARProcMod), water extraction using U-Net (WaterUNet), an accuracy assessment, and an analysis of the relationship between the water level and water area. The first step generates long-time-series Sentinel-1 SAR $\sigma^0$ images by using the SARProcMod model. Then, in the second step, a modified U-Net based water mapping model (WaterUNet) is proposed. To evaluate the performance of WaterUNet, some artificial samples are chosen for the confusion matrix and F1 score. Finally, we analyze the interannual and seasonal variation for the water body, as well as the relationship between the water area and the water level from Poyang gauging station.

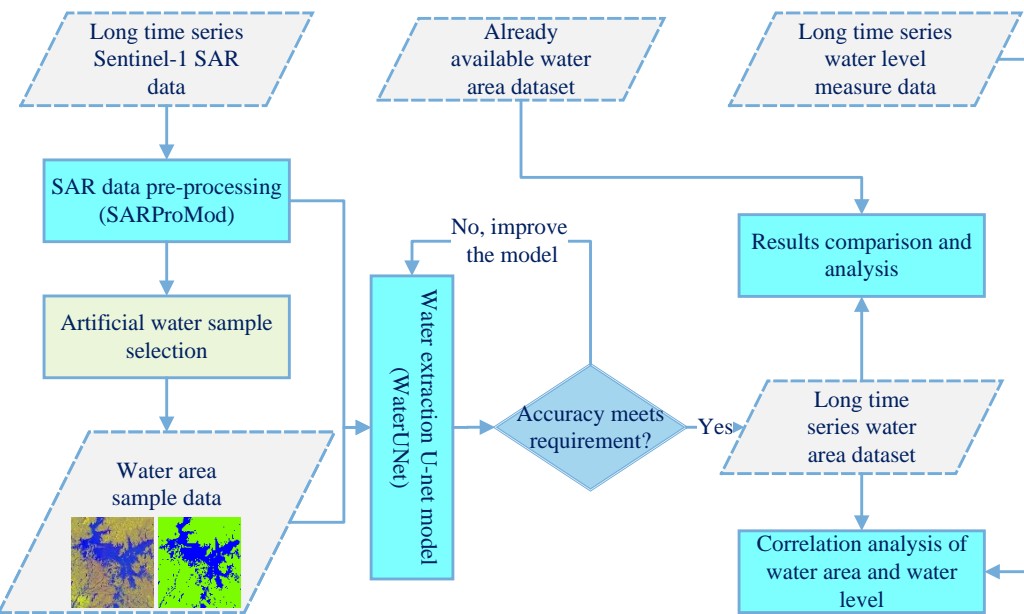

**Figure 5.** The flowchart showing the overall methods used in this study.

### 2.4. Data Pre-Processing

The European Space Agency (ESA) has developed the open-source software–SNAP (version: 8.0.0, https://sentinel.esa.int/web/sentinel/toolboxes/sentinel-1, accessed on 2 September 2021)—for Sentinel satellite data, which also provides the Python interface (SNAPPY) for secondary development. In order to batch process the 455 scenes of Sentinel-1 SAR data, we developed a processing module (SARProcMod) based on SNAPPY and Python to process all the data, which supports products of type SLC and GRD. Figure 6 shows the flowchart of how to pre-process the Sentinel-1 SAR data. Finally, the 10 m-resolution Sentinel-1 SAR dataset was produced for water extraction.

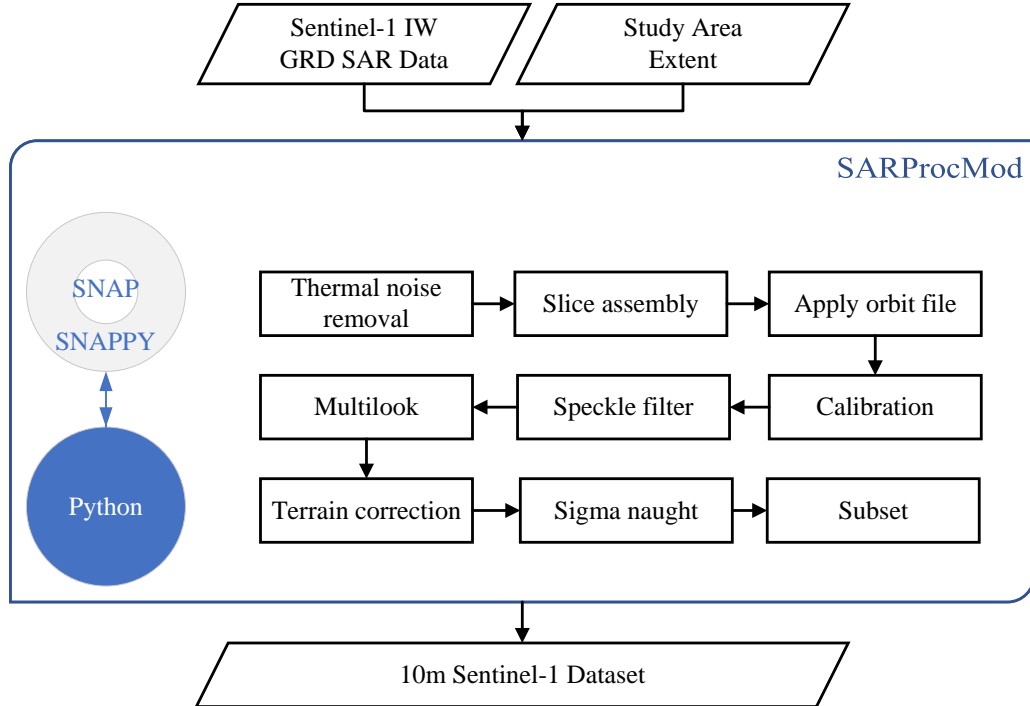

**Figure 6.** The flowchart for the SARProcMod for Sentinel-1 SAR pre-processing engine.

The workflow included nine steps. First, thermal noise removal was applied to all Sentinel-1 SAR data. Second, slice assembly was performed on the two consecutive slices. Third, the orbit files were applied to all Sentinel-1 SAR data to update the orbit state vectors in the abstract metadata. Fourth, radiometric calibration was applied to all Sentinel-1 SAR data to output radar backscatter bands ($\sigma^0$). Fifth, a Refined Lee filter was utilized to remove speckle noises. Sixth, multi-looking was applied to make the pixels square. Seventh, radar backscatter bands were orthorectified using the Range Doppler Terrain Correction algorithm with SRTM DEM data (with a spatial resolution of 1 s). Eighth, the backscatter coefficient (in dB) was acquired from the orthorectified radar backscatter band by the equation $10 \times \log_{10}\left(\sigma^0\right)$. Ninth, all the Sentinel-1 SAR data were clipped according to the extent of the study area. All the clipped images were processed by thresholding off values higher than 99.9% and lower than 0.01% from the histogram to remove the pixels with abnormal $\sigma^0$. Then, the images were scaled to [0, 1] according the equation below:

$$Image_{new} = \frac{Image_{ori} - minV}{maxV - minV} \tag{1}$$

where the $Image_{ori}$ and $Image_{new}$ refer to the SAR images before and after the scale and $maxV$ and $minV$ refer to values corresponding to the 99.99% value and 0.01% value in each polarization channel for the image.

*2.5. Water Body Mapping Using WaterUNet*

Since valid pre-trained U-Net cannot be migrated to SAR imagery, we modified the classic U-Net to extract water body using SAR data. The flowchart of our U-Net-based water extraction method is illustrated in Figure 7. There were two parts to this method: the encoding part (left part of Figure 7), which provides the basis for object category recognition, and the decoding part (right part in Figure 7), which provides the basis for accurate segmentation and positioning. Here, each convolution block had the same workflow, a $3 \times 3$ convolution layer with padding, a batch normalization layer, an activation layer (ReLU activator), and one more. The decoder was mainly composed of a deconvolution structure, a convolution block, and a concatenation. Finally, the deconvolution structure upsampled the feature map to achieve pixel-level classification. The gray arrows from left to right refer to the skip-connection, which can be used to fill in the low-level information to improve the segmentation accuracy. The loss function used in WaterUNet was *binary_crossentropy*, the learning algorithm was *Adam*, and the accuracy evaluation function was *accuracy*. All the algorithms are provided in the Google Tensorflow framework [65].

The samples used to train the model were pixel-level, meaning that each pixel in the sample was labeled with an attribute value (1 for water and 0 for non-water). Due to the medium resolution of the Sentinel-1 SAR images, the backscatter coefficient threshold segmentation method and direct visual interpretation method were used to initially classify the images; then, the GIS tools were used to post-process the misclassification and fragmentation patches in the results through manual discrimination to ensure a good internal connectivity, complete boundaries, and correct types of feature categories as far as possible [43].

Considering the resolution of the image, patches were firstly cropped to a $64 \times 64$ size using a sliding window in order to better capture the spatial features of water bodies for the model (Figure 8). To better learn the features of water bodies in SAR images and considering the sample class imbalance, samples with a water body ratio of 25% to 75% were selected as sample sets. The diversity and variability of the sample data were increased through data enhancement, including data rotation by 90°, 180°, and 270°. A total of 11,220 patches were obtained and used to build a more accurate water dataset. A total of 80% of the samples in the dataset were used for training and 20% were used for validation.

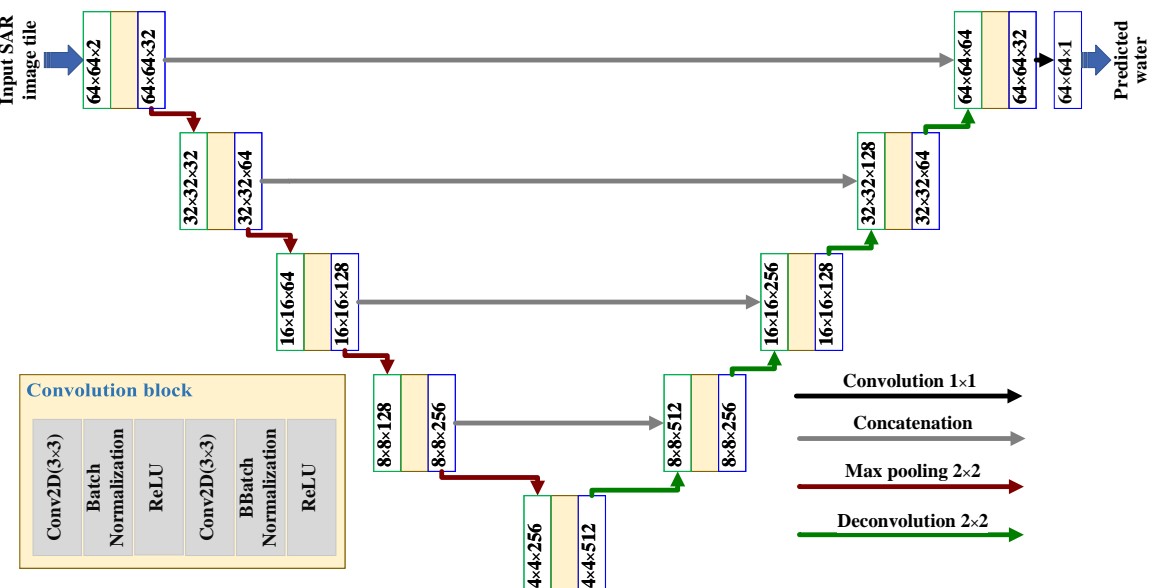

**Figure 7.** The WaterUNet structure for water extraction (total params: 7,771,585; trainable params: 7,765,697; non-trainable params: 5888) The numbers in the green box are for the input dimension of the convolution block and the numbers in the blue box are for the output dimension.

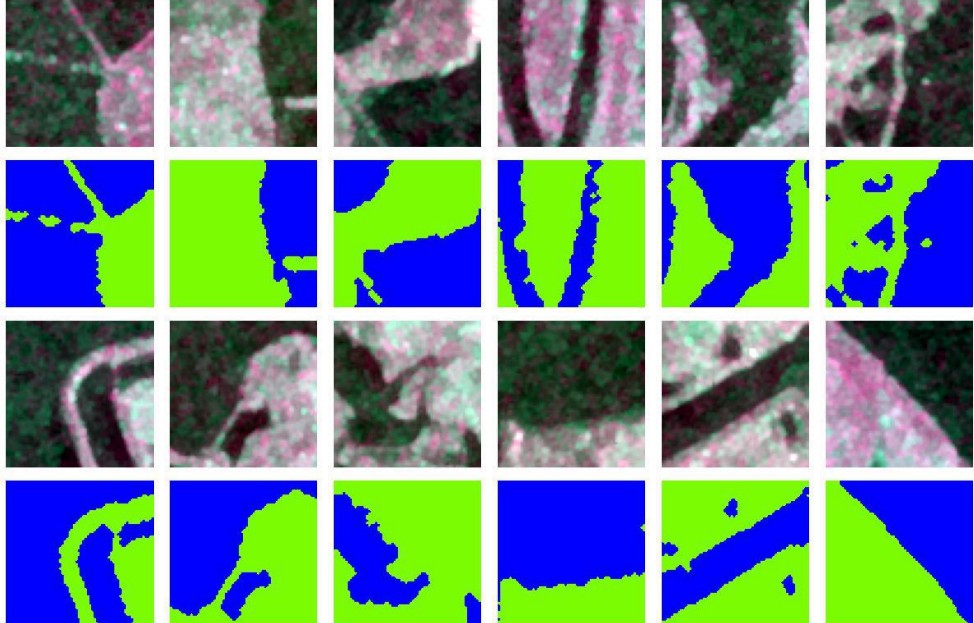

**Figure 8.** Examples of training data and validation data. Blue: water; green: other land types.

*2.6. Classification Accuracy Evaluation*

In this study, we used the confusion matrix (Table 1) and F1 score to assess the performance of our method. The confusion matrix [66] is a very popular measure used to solve classification problems. Confusion matrices represent counts from predicted and actual values. The output "TN" stands for True Negative, which shows the number of negative examples that were classified accurately. Similarly, "TP" stands for True Positive, which indicates the number of positive examples that were classified accurately. The term "FP" shows False Positive value—i.e., the number of actual negative examples classified as positive—while "FN" means a False Negative value, which is the number of actual positive examples classified as negative [66].

**Table 1.** Confusion matrix used for binary classification.

|  |  | Predicted | |
|  |  | Non-Water | Water |
| --- | --- | --- | --- |
| Actual | Non-water | TN | FP |
|  | Water | FN | TP |

The metrics most frequently used while performing classification are accuracy, precision, recall, Cohen's Kappa coefficient, and F1 score. The accuracy of an algorithm is represented as the ratio of correctly classified samples ($TP + TN$) to the total number of samples ($TP + TN + FP + FN$) [67].

$$Accuracy = \frac{TN + TP}{TN + FP + FN + TP} \tag{2}$$

The precision of an algorithm is represented as the ratio of samples correctly classified as positive ($TP$) to the total samples predicted to be positive ($TP + FP$) [67].

$$Precision = \frac{TP}{TP + FP} \tag{3}$$

The recall metric is defined as the ratio of samples correctly classified as positive ($TP$) divided by the total samples actual positive ($TP + FN$). Recall is also called as sensitivity [67]:

$$Recall = \frac{TP}{TP + FN} \tag{4}$$

Cohen's Kappa coefficient ($k$) [68] is a statistic that assesses the level of agreement between two classifiers on a classification problem. Cohen's Kappa coefficient can range from $-1$ to $+1$, where a value of 1 means that the agreement between the classifiers is perfect and a value of 0 indicates that any agreement is expected to be the result of random chance [68]. It is calculated by:

$$k = \frac{Accuracy - P_e}{1 - P_e} \tag{5}$$

where $P_e$ is the probability of random agreement and calculated by:

$$P_e = \frac{(TP + FN)(TP + FP) + (TN + FN)(TN + FP)}{(TP + TN + FP + FN)^2} \tag{6}$$

The *F1 score* is the harmonic mean of precision and recall, where an *F1 score* reaches its best value at 1 and worst at 0. It is more objective than overall accuracy in our binary classification case because a water body mostly covers a small portion of the image under evaluation [43,52]:

$$F1\ score = \frac{2 \times (Precision \times Recall)}{Precision + Recall} \tag{7}$$

## 3. Results

### 3.1. The Variation in $\sigma^0$ for Different Land Cover

After pre-processing all the Sentinel-1 SAR data, the series SAR images were utilized to investigate the multi-temporal backscatter coefficient properties (e.g., mean and standard deviation) for persistent water area (PW), permanent non-water area (NW), and seasonal water area (SW). Here, we selected all the Sentinel-1 SAR data for the years 2016 and 2017. The statistical results of $\sigma^0$ for each class are shown in Figure 9.

From Figure 9, we can see that the backscatter for the permanent water area is weaker but more stable than that for the non-water area, and the backscatter coefficients under VH polarization (around $-29$ dB) are lower than those under VV polarization (around $-23$ dB)

and much more concentrated, with a smaller variance (Figure 9b). For the seasonal water area, the backscatter is lower when the area is submerged by water. However, when the water body retreats, the backscatter becomes stronger. Therefore, there are two periods of strong backscatter in these two years, especially from August 2016 to April 2017. For the non-water area, the backscatter is strong and much more concentrated, with a smaller variance (Figure 9b). Similarly, the backscatter coefficients under VH polarization (around −15 dB) for the non-water area are lower than those under VV polarization (around −9 dB). We can also see that there is no overlap between the backscatter coefficients under different land covers and different levels of polarization, especially between water bodies and other types of land cover. Therefore, long-time-series Sentinel-1 SAR data can be used for water body extraction [31].

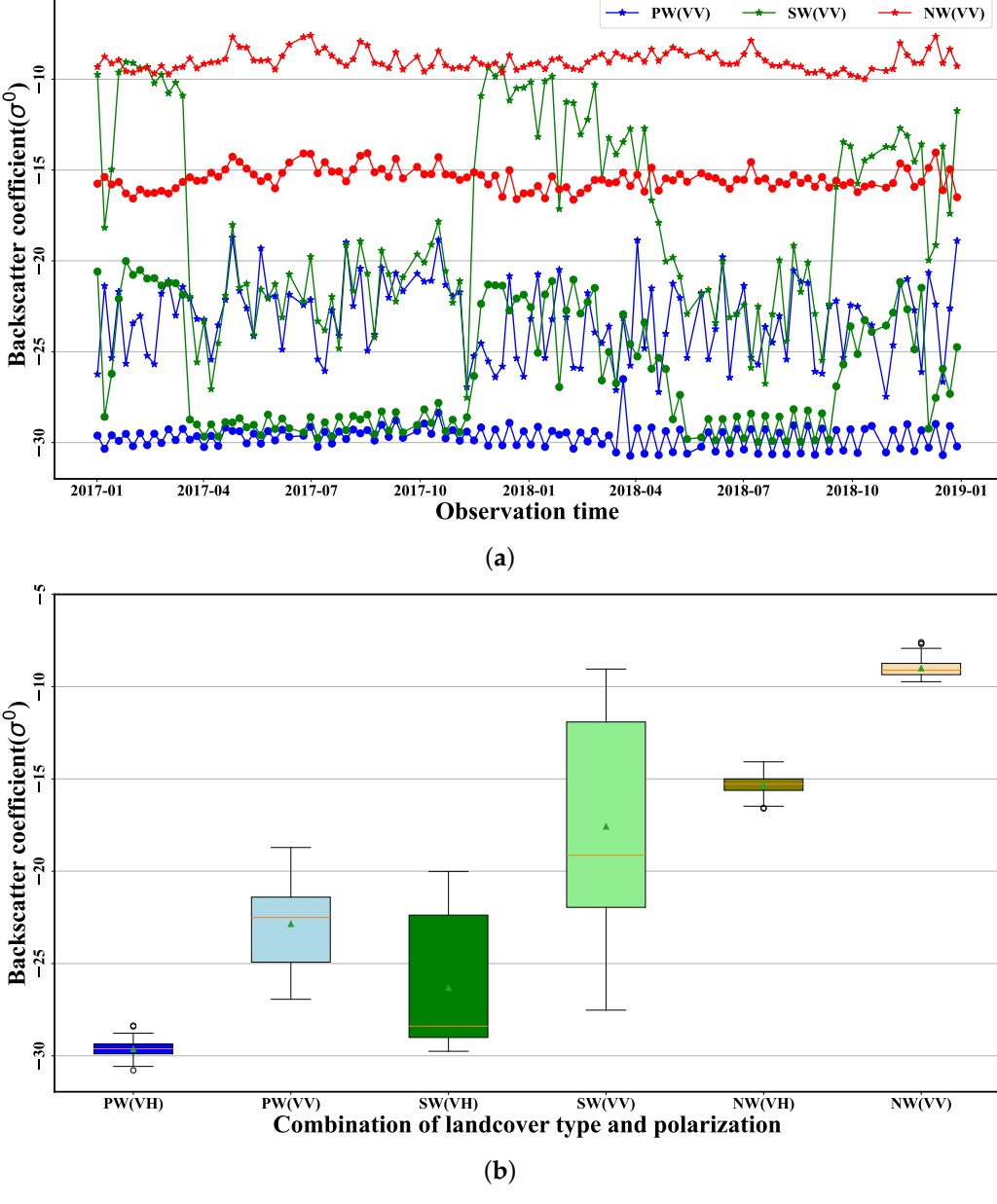

(a)

(b)

**Figure 9.** The temporal variations in radar backscatter coefficient from Sentinel-1 SAR data for different land cover types and polarization. PW is for persistent water area, SW is for seasonal water area, and NW is for non-water area.

### 3.2. Validation of the Model

WaterUNet was implemented under a Google Tensorflow framework and deployed on a Debian Linux workstation, with two Intel(R) Xeon(R) Gold 6134 CPUs and NVIDIA Quadro P5000 GPU. Due to the compact structure of WaterUNet (small size and fewer parameters), the training process achieved an almost optimal performance after 10 epochs with a high accuracy and small loss. The accuracy–loss curves of the training process are shown in Figure 10, where the accuracy curve was used to measure the performance of the model, while the loss curve was used to further optimize the model [43]. During the training process, the loss and accuracy values obtained for training samples were 0.98 and 0.03, while they were 0.98 and 0.03 for validation samples, which proved the validity of the WaterUNet's network structure.

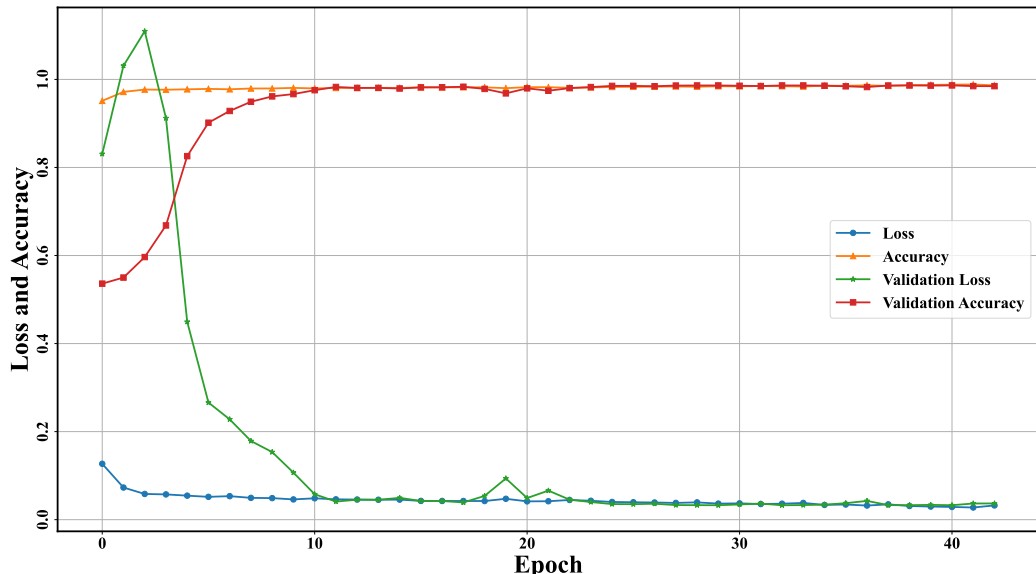

**Figure 10.** Accuracy–loss curves for WaterUNet with a batch size of 128.

Then, the performance of WaterUNet was evaluated using the testing samples and the samples manually and randomly selected from the two scenes of SAR data acquired on 11 July 2015, and 21 April 2019. The confusion matrix and corresponding classification accuracy parameters are as shown in Table 2.

**Table 2.** Accuracy assessment for WaterUNet classification results.

| Validation Dataset | Kappa Coefficient | Accuracy | Precision | F1 Score |
|---|---|---|---|---|
| Testing samples | 0.96 | 0.98 | 0.98 | 0.98 |
| 11 July 2015 | 0.98 | 0.99 | 1.0 | 0.99 |
| 21 April 2019 | 0.97 | 0.98 | 0.99 | 0.98 |
| Average | 0.97 | 0.98 | 0.99 | 0.98 |

From Table 2, it can be seen that the Kappa coefficient, accuracy, precision, and F1-scores for WaterUNet are 0.97, 0.98, 0.99, and 0.98, respectively, indicating the superior classification accuracy of the proposed method. Based on the validation of the effectiveness of WaterUNet, we applied the trained model for all the data in the 10 m Sentinel-1 SAR dataset.

### 3.3. Annual and Interannual Variation of Water Area

After pre-processing all the Sentinel-1 SAR data, we used the trained WaterUNet to extract water body data and then obtained the water body distribution maps of Poyang

Lake corresponding to all 324 SAR observations. Poyang Lake, as previously stated, has distinct wet and dry seasons. During the wet season, each small lake is connected to the main lake and water covers most of the study area. Due to the severe flood that took place in Poyang Lake basin on 20 July 2020, the water area reached its highest level for the past 6 years (2714.08 km$^2$), covering 96.4% of the study area (Figure 11a). The water surface shrank throughout the dry season, and the small lakes were cut off from the main lake. On 4 January 2020, the water area reached its lowest value in the past 6 years (634.44 km$^2$), covering 22.5% of the study area (Figure 11b).

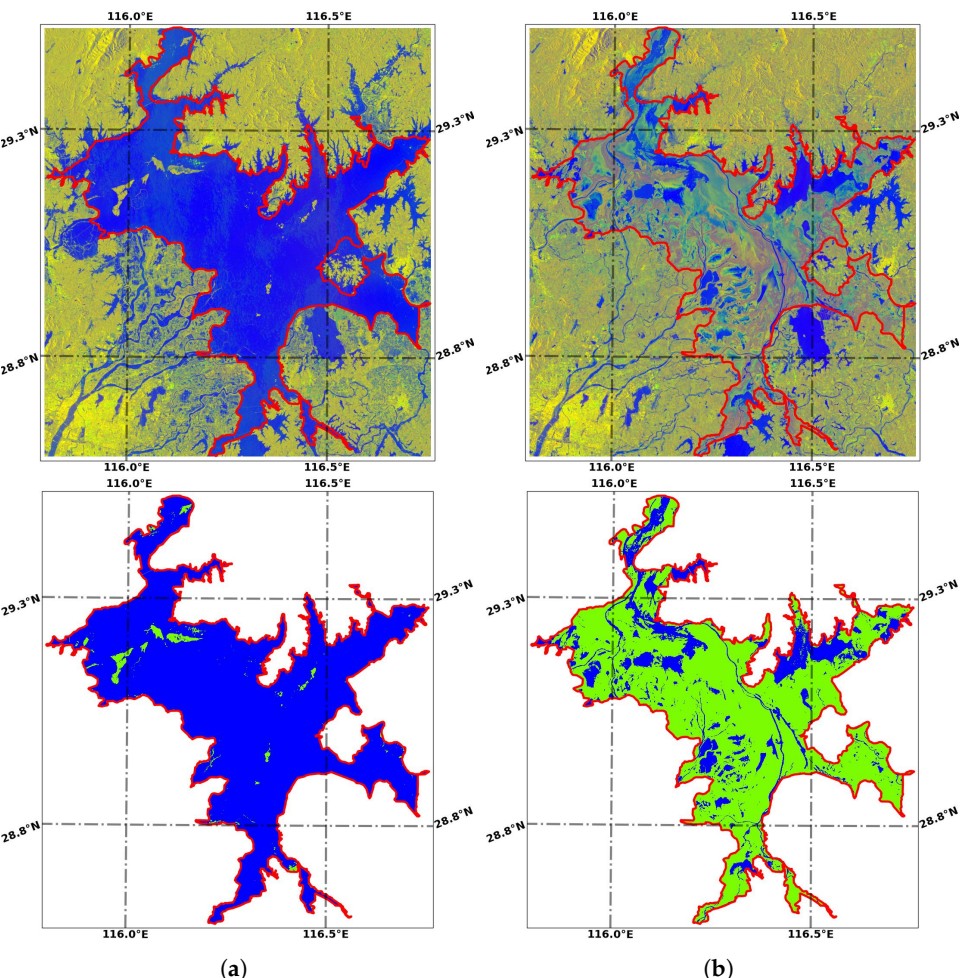

(a)                                   (b)

**Figure 11.** The maximum and minimum water area from 2014 to 2021. Blue is for water and green is for non-water. (**a**) maximum water area on 20 July 2020; (**b**) minimum water area on 4 January 2020.

After obtaining the water body distribution maps corresponding to all 324 SAR observations, Equation (8) was used to calculate the inundation frequency (IF) for Poyang Lake. The spatial distribution of the IF factor can be used to reflect the spatial–temporal distribution features of water bodies. The longer the water body lasts, the greater the IF will be. The larger the water body extent is, the greater the IF will be.

$$Inundation \ frequency = \frac{Water \ Observation \ Times}{Total \ Observation \ Times} \times 100\% \qquad (8)$$

Figure 12 shows the IF distribution map corresponding to all the SAR data for our study area. From Figure 12, we can see that most of the study area is covered by seasonal water body area (areas with green color), becoming inundated by water and turning into mudflats and grasslands. The second larger part of the study area is a perennial water

area (areas with blue color, and an IF greater than 0.75) that is covered by water almost all year round. The IF for some of the perennial water area is close to 1.0, meaning that these areas are always covered by water all year round. Some study areas are non-water areas (areas that are bright green with an IF close to 0.0), such as the small islands in Poyang Lake. According to the shape of perennial water bodies, water tanks and water transfer channels that link tanks inside the lake system can be identified. The spatial distributions of IF also highlight the different storage areas according to the duration of the water presence and their geographical location within the hydrological system. Some of the study area is mudflats (areas with orange color and an IF between 0.5 and 0.75). The lower the terrain is, the more likely it is to be flooded, and the IF distribution also reflects the topography of the lake basin [69]. Figure 12 shows that the IF generally varied along the gradient of the elevation, exhibiting maximum values within the deep flow channels and permanently flooded sub-lakes [59].

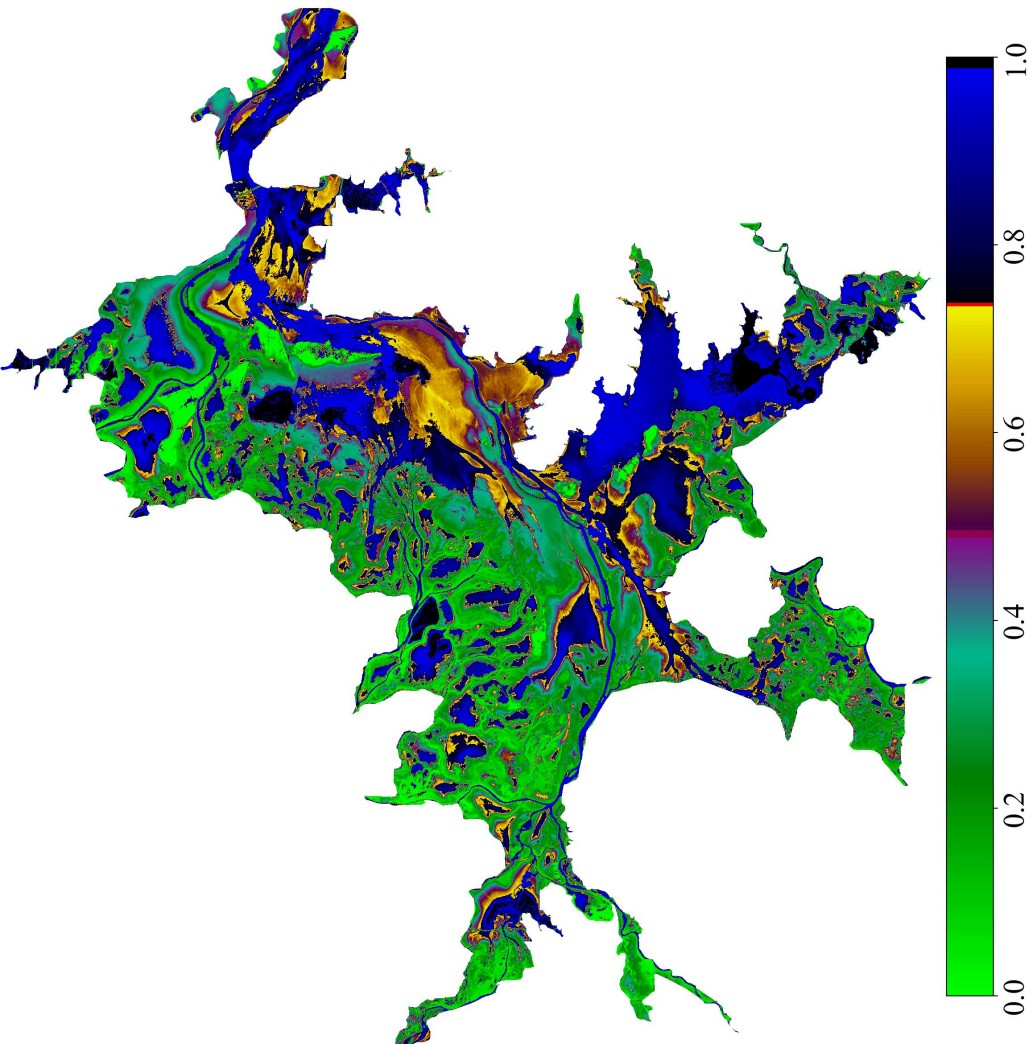

**Figure 12.** The inundation frequency distribution map of Poyang Lake generated from all the 324 SAR observations during 2014–2021.

We conducted an annual statistical evaluation to study the yearly variation in the IF. In the years 2014 and 2021, the SAR data were acquired in the dry season, resulting in a lower IF for the study area. Conversely, in the year 2015, most of the SAR data were acquired in the wet season, resulting in a higher IF for the study area. Since the data for 2014, 2015, and 2021 are not available throughout the year, the IF distribution maps for these years are not so meaningful. Thus, here we focus on the analysis of the IF from 2016 to 2020 (Figure 13).

From Figure 13, we can see that the IF values in 2020 are generally the highest, followed by those in 2016, 2017, 2019, and 2018, which is consistent with the order of the duration of a high water level. Due to the longer duration of the water level higher than 15 m, most of the IF values in 2020 are higher than 0.2.

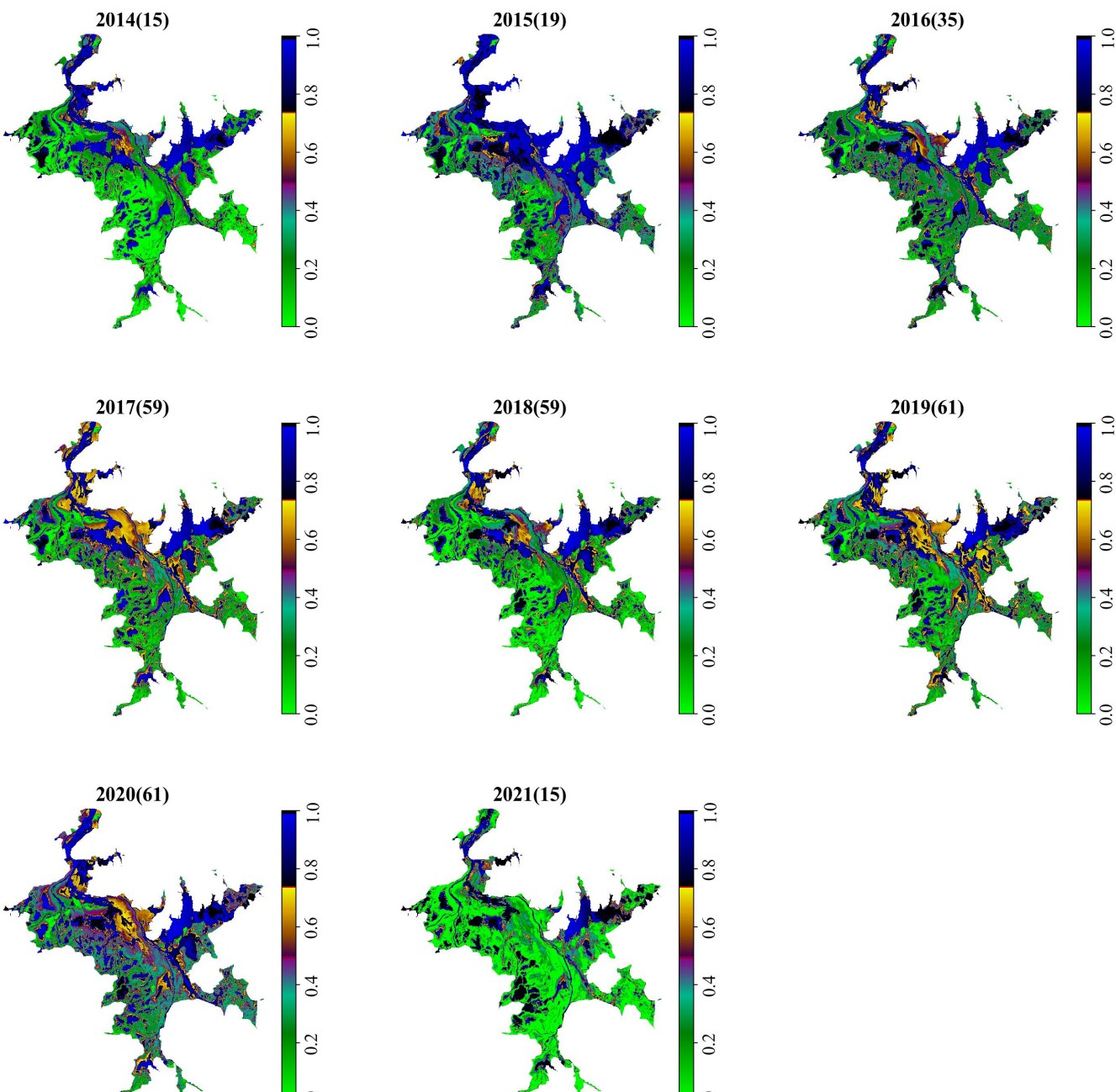

**Figure 13.** The IF distribution maps for different years from 2014 to 2021. The numbers in the brackets are for the observation times in that year.

The monthly statistics from the IF were obtained to study the interannual variation in the IF. Figure 14 shows the IF distribution maps for our study area in different months. The missing SAR data in January, February, and March 2015 did not influence the IF in these months. From Figure 14, we can see that, as Poyang Lake entered the wet season in April, the monthly IF of the study area increased until October. As Poyang Lake entered the dry season, the monthly IF began to decrease, which lasted until March of the next

year. The central area of the lake (orange area in Figure 14 for September) suffered the most pronounced cyclical change, cycling back and forth between inundation and un-inundation.

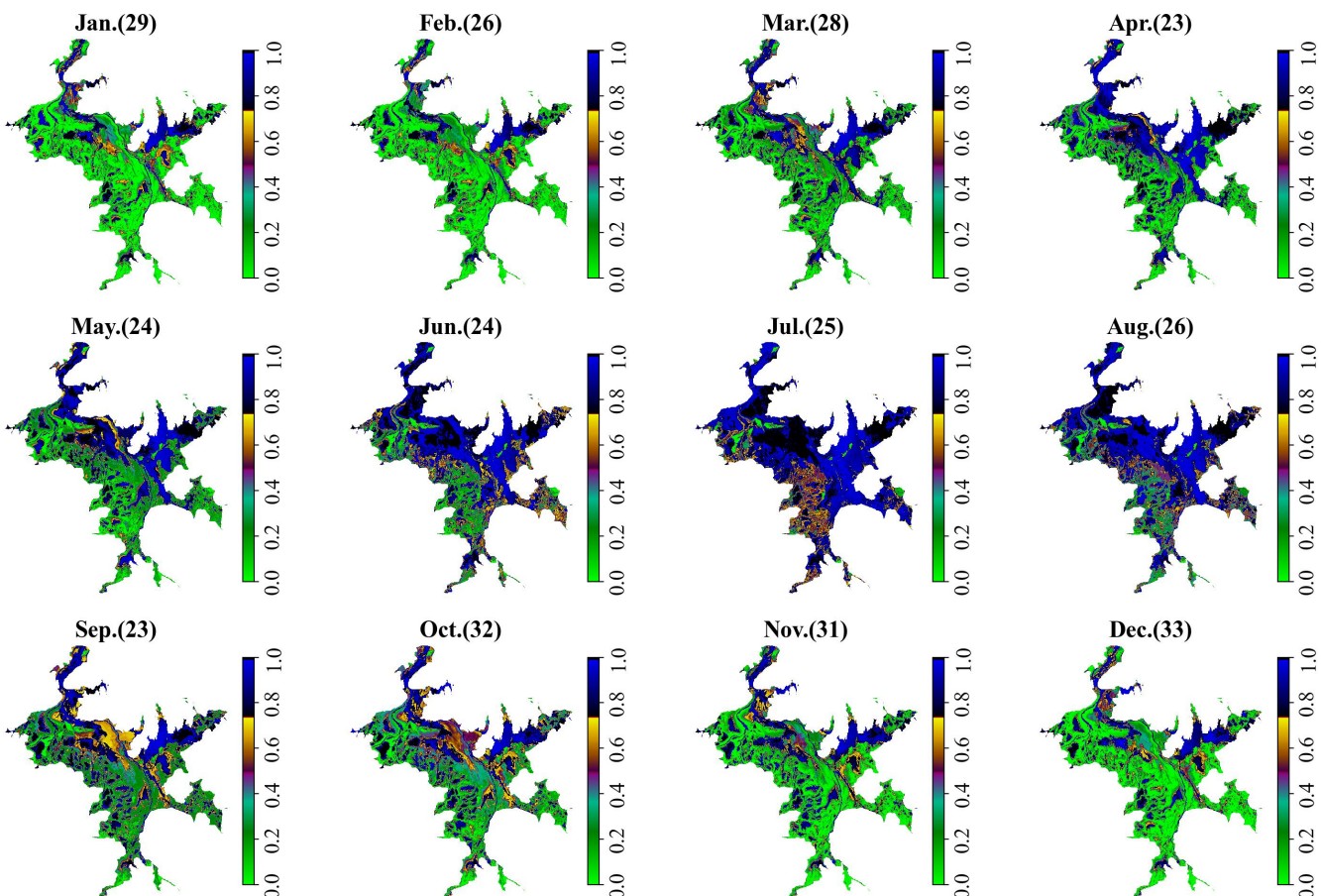

**Figure 14.** The IF distribution maps for different months from 2014 to 2021. The numbers in the brackets are the observation times for that month across all years.

### 3.4. The Relationship between Water Level and Water Area

Figure 15 shows the interannual and seasonal fluctuations in the water area obtained using WaterUNet and the histogram for Poyang Lake from October 2014 to March 2021. Similar to the water level, the water area also fluctuated regularly from 2014 to 2021. The similar fluctuation pattern of the water area can indirectly prove that a 5~7-day revisit cycle is sufficient for monitoring water body changes for Poyang Lake. Figure 15 also shows that the highest water area occurred in July every year as the water level reached the highest level. Under the combined action of Yangtze River flood and the five upstream rivers flood, the water level of Poyang Lake was always at a high water level, resulting in the longest duration when the water area was higher than 2500 km$^2$. Although the water level in 2020 was higher than that in 2016, the water area in 2020 was not larger because the water level had exceeded the warning level and was also affected by the scope of the study area.

After obtaining both the water area and water level data, we chose 284 pairs of water area and water level data acquired on the same day and analyzed the relationship between them (Figure 16). Firstly, we calculated the correlation coefficient between the water area and water level, which was up to 0.92, showing a good correlation between them. Then, the quadratic polynomial fitting equation was obtained as $y = -21.796x^2 + 975.334x - 8150.724$, and the fitting $R^2$ reached 0.88. The quadratic polynomial can sufficiently express the relationship between the water level and water area; thus, the expression can be used to estimate the water area or water level of Poyang Lake when the water level or water area data are invalid. From Figure 16, we can also see that, when the water level reaches over 20 m, the water area does not increase significantly due to the dikes around Poyang Lake [62] and

the limitation of the extent of the study area. As shown in Figure 1, Poyang gauging station is located on the east bank of Poyang Lake, so the relationship can be improved by using the water level from the gauging station located in the central part of Poyang Lake.

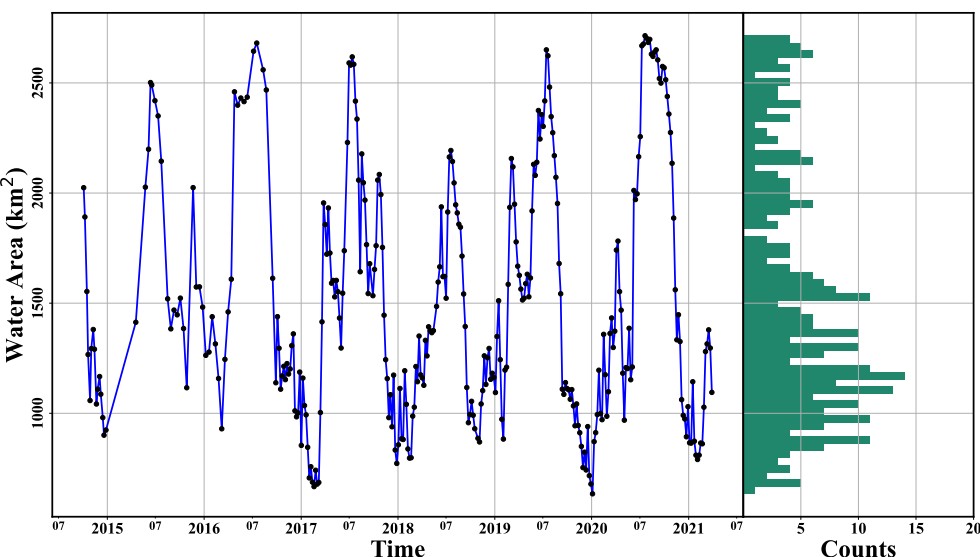

**Figure 15.** The water area series obtained from Sentinel-1 SAR data using WaterUNet for Poyang Lake from 2014 to 2021 and its histogram. The black dots are for the water area, and the blue line is for the trend of water area. The green bars are for the count in each water area interval (64 bins).

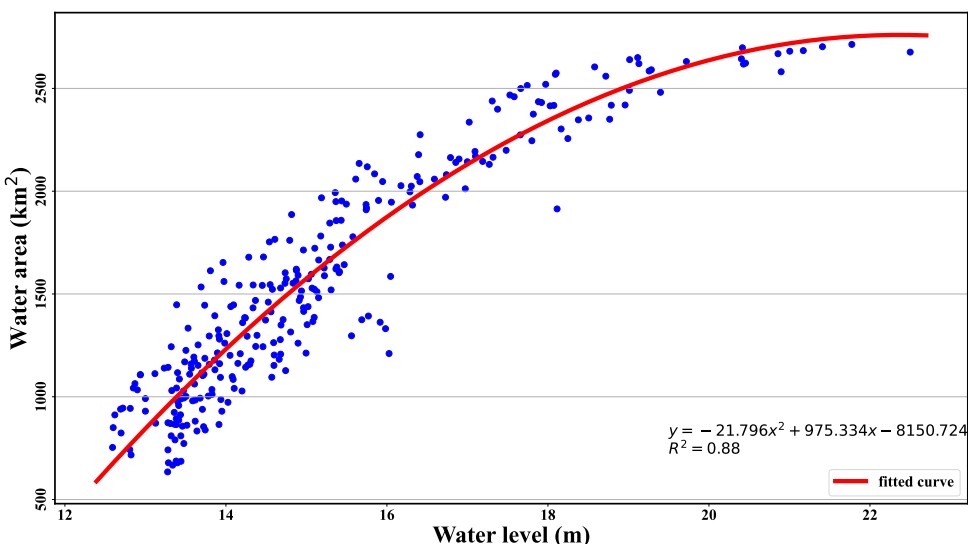

**Figure 16.** The relationship between the water level and water area for Poyang Lake. The red line is for the fitted curve, with "*y*" for the water area and "*x*" for the water level.

## 4. Discussion

### 4.1. Analysis of the Influence Factors

Although the test accuracy of WaterUNet is satisfactory, there are still some factors that influence its classification accuracy. Firstly, all the Sentinel-1 SAR data used in this work were used in Interferometric Wide Swath (IW) Mode [70], which allows a large swath width (250 km) and can cover Poyang Lake with one or two consecutive scenes. The IW mode images three sub-swaths using Terrain Observation with Progressive Scans SAR (TOPSAR) [70], and Poyang Lake is in the middle and far sub-swaths. Although the TOPSAR technique ensures homogeneous image quality throughout the swath, the backscatters near the edge of the sub-swaths are different and more pronounced in the VH

polarization images (the blue box in Figure 17b), which can be observed in most data and may lead to the misclassification of water bodies (Figure 17). In addition, the scalloping effects in the Sentinel-1 TOPSAR SAR data will also affect the backscatter in radiometric quality in the order of 1-2 dB, especially in VH polarization (the ripples are shown in the yellow and green boxes in Figure 17b) [71,72].

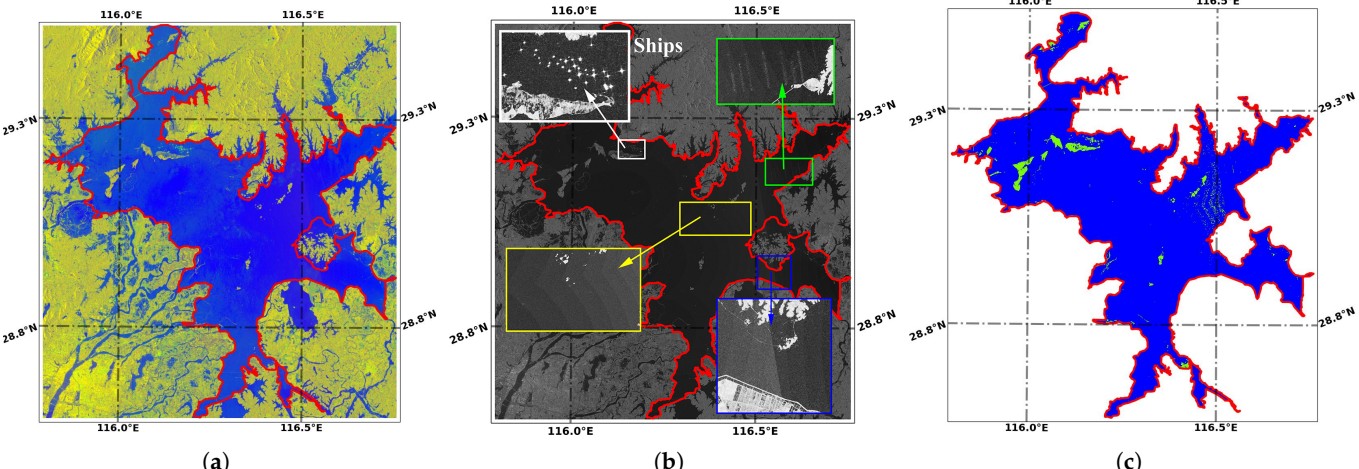

**Figure 17.** The influences of subswath and scalloping effects on water extraction, taking the data acquired on 14 July 2020, as examples. The blue box in (**b**) shows the enhanced subswath edge, while the green and yellow box show the scalloping effect. The corresponding area in (**c**) shows water misclassified as non-water, with blue for water and green for non-water. (**a**) Sentinel-1 SAR data in VH/VV/(VH + VV); (**b**) Sentinel-1 SAR data in VH; (**c**) classification result.

Another known problem in the use of SAR data for water body mapping is the floats and sediments. When the upstream flood flows into Poyang Lake, the floats or sediments will cause a change in the water surface's backscatter, especially in VH polarization. Figure 18 shows the Sentinel-1 SAR data acquired on 27 May 2020; 2 June 2020; and 8 June 2020, at the water levels of 14.6 m, 16.0 m, and 16.9 m. From 27 May 2020, the water level began to rise dramatically (red dots in Figure 2) due to the flow caused by upstream precipitation, which brought many floats and sediments, resulting in the change of the backscattering from specular reflection to Bragg scattering. Consequently, in the data acquired on 2 June, there were large areas where the backscatter was different from that of water surface due to the floats and sediments, leading to the misclassification of areas as non-water and resulting the omission of water bodies (Figure 18b). In addition, the boats on the lake will also lead to omission due to the strong backscatter caused by the hull's metal material (Figure 17b).

Reschke et al. [73] mentioned the restraining factor of weather conditions (rain and wind) in C-band ENVISAT ASAR data. Wind-induced waves also affect backscatter for C-band Sentinel-1 SAR data [31] and cause the misclassification of water bodies. Figure 19 shows the SAR data acquired on 15 February 2020, when it was raining or snowing with a level 5 north wind around the lake (https://lishi.tianqi.com/, accessed on 12 December 2021), Ducang County, and Poyang County, maybe with stronger wind over the lake surface. The stronger wind caused waves on the lake surface, which strengthens the backscatter from the water surface by altering the backscattering from specular reflection to Bragg scattering [31] and cause misclassification.

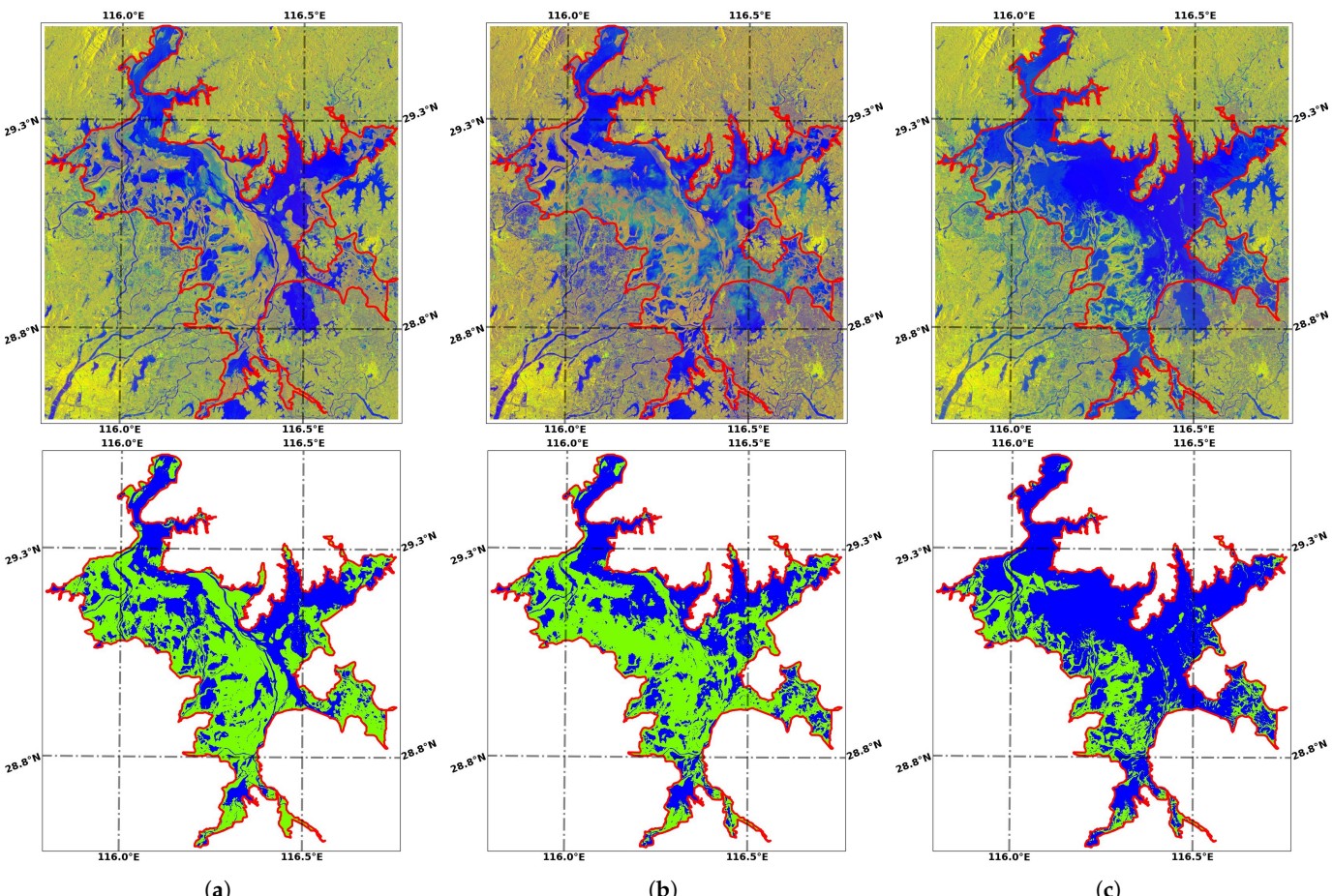

**Figure 18.** The influences of floats and sediments on water extraction. The upper figures are for the Sentinel-1 SAR data displayed in VH/VV/(VH + VV) and the lower figures are for the classification results, with blue for water and green for non-water. (**a**) 27 May 2020, water level of 14.6 m; (**b**) 2 June 2020, water level of 16.0 m; (**c**) 8 June 2020, water level of 16.9 m.

Input data transformation has been recognized as an impactful pre-processing task when using machine learning algorithms such as ANN models. Accordingly, the remote sensing data of this study were transformed using the histogram threshold scaler and the maximum–minimum scaler approach. Ghorbanian et al. [74] proved the capability of the Adam learning algorithm in handling untransformed input data, which were also employed in WaterUNet.

Although these factors affect the classification accuracy, WaterUNet still extracts water body data with a high accuracy, which also proves the robustness of WaterUNet.

### 4.2. Comparisons with Other Studies

As the largest freshwater lake in China, Poyang Lake has been paid more and more attention, especially its water body changes. In previous studies, optical remote sensing data such as MODIS data and Landsat data were the primary data sources for mapping Poyang Lake water body [61,75–77], mainly due to the free policy and long-term availability of these data. With the temporal resolution of 16 days, only 8 cloud–free Landsat TM/ETM+ data were used to retrieve the water body for Poyang Lake from November 1999 to October 2000, mainly due to the influences of clouds [75]. Similarly, the influences of clouds resulted in only 466 cloud–free MODIS products available to investigate the spatial–temporal distribution of inundation in Poyang Lake during 2000–2011 [76]. Meanwhile, SAR data were used to study water body mapping methods [2,13,78], and there were few long-term water body mapping based on SAR data due to its high price. In the past few years,

Sentinel-1 SAR cloud–free data have become another data source for Poyang Lake water body mapping as the high spatial–temporal resolution [1,32,37]. All available Sentinel-1 SAR data in GEE [1], 71 Sentinel-1A data (41 observations, 12 days temporal resolution) [37] and 15 Sentinel-1A SAR data (12 days temporal resolution) were used for Poyang Lake water body mapping or flooding monitoring [32]. Here, we collected 455 available Sentinel-1A/B SAR data (324 observations) for Poyang Lake water body mapping and constructed a consecutive high spatial (10 m resolution) and temporal (6 days temporal resolution) water body dataset, which can reflect the water body changes well especially in the flood season.

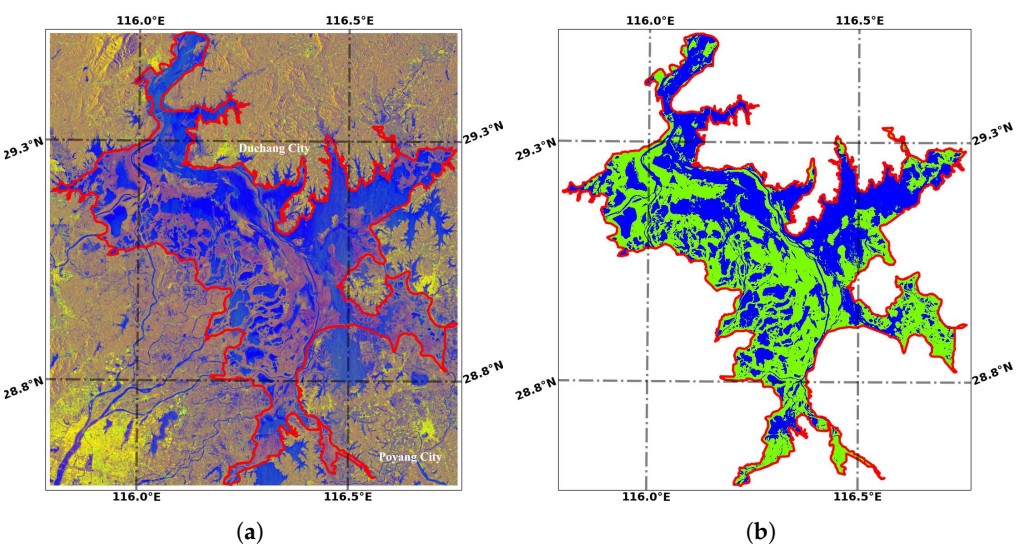

(a)  (b)

**Figure 19.** The influence of wind-induced waves on water extraction. (**a**) is for the Sentinel-1 SAR data acquired on 15 February 2020, and displayed in VH/VV/(VV + VH), and (**b**) is for the classification result, with blue used for water and green for non-water. (**a**) Sentinel-1 SAR data; (**b**) classification result.

The high spatial-temporal water body dataset in China (HSWDC) was constructed based on Sentinel-1 SAR data and GEE platform, which provided the monthly water dynamics of Poyang Lake [1]. HSWDC was construed using the backscatter threshold segmentation method, where pixels with $\sigma_{VV} \leq -15$ dB and $\sigma_{VH} \leq -23$ dB were regarded as water body, and a high classification accuracy (overall accuracy: 0.93, Kappa coefficient: 0.86) was achieved. Sentinel-1A water index (SWI) and the SWI threshold classification method were used to map Poyang Lake water body with the threshold of 0.2, and the overall accuracy and Kappa coefficient were 96.52%, 0.8981; 96.16%, 0.9102 for Poyang Lake in 27 September 2016 and 9 September 2015 [37]. However, the overall accuracy for OSTU and bimodal threshold segmentation (BTS) methods in [32] were 0.698 and 0.766, respectively. Thus, the threshold segmentation methods are efficient enough to map water body and easily deployed in GEE platform but depend only on the variance without considering the human prior knowledge [32] and the spatial relations between the pixel and its neighbors. The overall accuracy for HRNet, DenseNet121, SegNet, ResNet101, and DeepLab v3+ were 0.970 (highest precision and efficiency), 0967, 0.969, 0.966, and 0.956, which demonstrated that the CNNs appear to significantly outperform the traditional methods [32]. WaterUNet, based on U-Net, can take into account the human prior knowledge, the spatial relations [54] and learn high–level context features and then improve water body mapping. WaterUNet has the same structure as the conventional U-Net, but with 50% of convolution kernels for each layer and the reduced number of model parameters. The results show that WaterUNet can map water body with high accuracy (overall accuracy: 0.98, Kappa coefficient: 0.97) and a good generalization ability. Recent studies have also optimized the CNN-based water body mapping models to improve the accuracy [46,51,54]. The drawbacks of CNN-based models are the need for a large number of training samples and its time-consuming quality

during the training process. However, when the model is trained, it will take a short time to make predictions from new images [32].

Some studies have also provided IF maps in their works, and most of these maps were generated from MODIS, Landsat optical data [61,75–77], ENVISAT ASAR data [13], ENVISAT Low and Medium Resolution data [78], or the simulation of hydrodynamic models [59]. Although the data sources are different, the IF from different studies have similar temporal-spatial distribution trend, with IF generally varying along the gradient of the elevation, exhibiting maximum values within the deep flow channels and permanently flooded sub-lakes. In addition, many more details are presented in the the proposed 10 m resolution IF distribution map. Some researchers have also studied the relationship between the water area and water level of Poyang Lake. The relationship during 2000–2020 was provided in [61] as $water\ level = 0.003 \times water\ area + 5.494$ with $R^2 = 0.926$, where the water level data were from DAHITI [79]. The relationship during 2000–2015 was provided in [62] as $water\ area = 231.01 \times water\ level - 722.04$ ($water\ level < 15.0\ \text{m}$) and $water\ area = 38.493 \times water\ level + 2387.9$ ($water\ level \geq 15.0\ \text{m}$), where the water level data were from Xingzi gauging station. The differences among these IF distribution maps and these relationships may be caused by the study periods, data sources, and the bathymetry of Poyang Lake [80]. The bathymetric changes in Poyang Lake were attributed to the combined effect of human activities (e.g., sand dredging [62,80–82], levee construction [62,81], and the impoundment of the Three-Gorges Dam [81]) and weather events (the excessive precipitation resulting in the increased bottom elevation [81] caused by the sediments).

## 5. Conclusions

Timely water monitoring, especially flood dynamic monitoring, is very important for water resource regulation, disaster assessment, and disaster mitigation. Due to its high temporal-spatial resolution, Sentinel-1 SAR is very suitable for detecting water as well as emergency flood mapping. As the largest freshwater lake in China, Poyang Lake's water area variation can reveal the occurrence of flooding in the upstream area of its lake basin and the Yangtze River. In order to monitor the change in the water area, we collected 455 scenes of Sentinel-1 SAR data from 2014 to 2021.

In this study, we provide a multi-temporal water body extraction framework for Sentinel-1 SAR data, and the SNAPPY and Google Tensorflow frameworks used in the proposed framework make it easy to implement and transfer at the code level. The data clipping method enhances the proposed framework. Basically, this framework relies on the scattering characteristics of water bodies, and the stability of water body scattering also ensures that it is applicable in other lakes. The preliminary result shows that the WaterUNet model realized accurate water extraction, with an F1 score of up to 0.99. The water area variation trend shows that Poyang Lake enters a flood period in July. In 2015, 2016, 2017, 2019, and 2020, the water areas in the study area were over 2500 km$^2$, and they reached 2714.08 km$^2$ in 2020. The IF distribution map for the Sentinel-1 SAR data clearly shows the seasonal water area, perennial water area, and non-water area, as well as the water tanks and water transfer channels in Poyang Lake. The IF distribution maps for different years and months show the interannual and annual variation in the water area. The IF distribution map for 2020 shows that a severe flood occurred in July, and the IF distribution maps for different months show the obvious wet season that occurs from April to September. The quadratic polynomial fitting results show that there is a good correlation between the water level from Poyang gauging station and the water area, with an $R^2$ of up to 0.88, which can be used to estimate the water area or the water level of Poyang Lake.

Due to the large amount of Sentinel-1 SAR data, data pre-processing takes a long time. Thus, in the future, we will try to transfer our framework to the GEE platform to reduce the pre-processing time for the Sentinel-1 SAR data and provide rapid emergency response services.

**Author Contributions:** G.S. proposed and designed the technique roadmap, contributed to the creation of the algorithmic methods, and collected the image data. W.F. proposed the technique roadmap and helped with the results analysis. H.G. and J.L. helped with the results analysis and reviewed the manuscript. All authors have read and agreed to the published version of the manuscript.

**Funding:** This work was funded by the Strategic Priority Research Program of the Chinese Academy of Sciences (Grant No. XDA19070102).

**Institutional Review Board Statement:** Not applicable.

**Informed Consent Statement:** Not applicable.

**Data Availability Statement:** The Sentinel-1 SAR data listed in Table A1 are available in https:// sci-hub.copernicus.eu/ (accessed on 18 April 2021). The other data presented in this study are available on request from the corresponding author.

**Acknowledgments:** The authors are grateful to ESA for providing the Sentinel-1 SAR data and http://www.jxssw.gov.cn/ (accessed on 1 July 2021) website for providing the water level data. We would like to express thanks to the anonymous reviewers for their voluntary work and the constructive comments to improve this manuscript.

**Conflicts of Interest:** The authors declare no conflict of interest.

## Appendix A

**Table A1.** The observation date and times for different years in this study.

| Year | Date | Times |
|------|------|-------|
| 2014 | 2014/10/3 2014/10/8 2014/10/15 2014/10/20 2014/10/27 2014/11/1 2014/11/8 2014/11/13 2014/11/20 2014/11/25 2014/12/2 2014/12/7 2014/12/14 2014/12/19 2014/12/26 | 15 |
| 2015 | 2015/4/18 2015/5/24 2015/6/5 2015/6/12 2015/6/17 2015/6/29 2015/7/11 2015/7/23 2015/8/16 2015/8/28 2015/9/9 2015/9/21 2015/10/3 2015/10/15 2015/10/27 2015/11/20 2015/12/2 2015/12/14 2015/12/26 | 19 |
| 2016 | 2016/1/7 2016/1/19 2016/1/31 2016/2/12 2016/2/24 2016/3/7 2016/3/19 2016/3/31 2016/4/12 2016/4/24 2016/5/6 2016/5/18 2016/5/30 2016/6/11 2016/7/5 2016/7/17 2016/8/10 2016/8/22 2016/9/15 2016/9/27 2016/10/3 2016/10/9 2016/10/15 2016/10/21 2016/10/27 2016/11/2 2016/11/8 2016/11/14 2016/11/20 2016/11/26 2016/12/2 2016/12/8 2016/12/14 2016/12/20 2016/12/26 | 35 |
| 2017 | 2017/1/1 2017/1/7 2017/1/13 2017/1/19 2017/1/25 2017/1/31 2017/2/6 2017/2/12 2017/2/18 2017/2/24 2017/3/2 2017/3/8 2017/3/14 2017/3/20 2017/3/26 2017/4/1 2017/4/7 2017/4/13 2017/4/19 2017/4/25 2017/5/1 2017/5/7 2017/5/13 2017/5/19 2017/5/25 2017/5/31 2017/6/6 2017/6/12 2017/6/24 2017/6/30 2017/7/6 2017/7/12 2017/7/18 2017/7/24 2017/7/30 2017/8/5 2017/8/11 2017/8/17 2017/8/23 2017/8/29 2017/9/4 2017/9/10 2017/9/16 2017/9/28 2017/10/4 2017/10/10 2017/10/16 2017/10/22 2017/10/28 2017/11/3 2017/11/9 2017/11/15 2017/11/21 2017/11/27 2017/12/3 2017/12/9 2017/12/15 2017/12/21 2017/12/27 | 59 |
| 2018 | 2018/1/2 2018/1/8 2018/1/14 2018/1/20 2018/1/26 2018/2/1 2018/2/7 2018/2/13 2018/2/19 2018/2/25 2018/3/3 2018/3/9 2018/3/15 2018/3/21 2018/3/27 2018/4/2 2018/4/8 2018/4/14 2018/4/20 2018/4/26 2018/5/2 2018/5/8 2018/5/14 2018/5/26 2018/6/1 2018/6/7 2018/6/13 2018/6/19 2018/6/25 2018/7/1 2018/7/7 2018/7/13 2018/7/19 2018/7/25 2018/7/31 2018/8/6 2018/8/12 2018/8/18 2018/8/24 2018/8/30 2018/9/5 2018/9/11 2018/9/17 2018/9/23 2018/9/29 2018/10/5 2018/10/11 2018/10/17 2018/10/29 2018/11/4 2018/11/10 2018/11/16 2018/11/22 2018/11/28 2018/12/4 2018/12/10 2018/12/16 2018/12/22 2018/12/28 | 59 |
| 2019 | 2019/1/3 2019/1/9 2019/1/15 2019/1/21 2019/1/27 2019/2/2 2019/2/8 2019/2/14 2019/2/20 2019/2/26 2019/3/4 2019/3/10 2019/3/16 2019/3/22 2019/3/28 2019/4/3 2019/4/9 2019/4/15 2019/4/21 2019/4/27 2019/5/3 2019/5/9 2019/5/15 2019/5/21 2019/5/27 2019/6/2 2019/6/8 2019/6/14 2019/6/20 2019/6/26 2019/7/2 2019/7/8 2019/7/14 2019/7/20 2019/7/26 2019/8/1 2019/8/7 2019/8/13 2019/8/19 2019/8/25 2019/8/31 2019/9/6 2019/9/12 2019/9/18 2019/9/24 2019/9/30 2019/10/6 2019/10/12 2019/10/18 2019/10/24 2019/10/30 2019/11/5 2019/11/11 2019/11/17 2019/11/23 2019/11/29 2019/12/5 2019/12/11 2019/12/17 2019/12/23 2019/12/29 | 61 |

**Table A1.** *Cont.*

| Year | Date | Times |
|------|------|-------|
| 2020 | 2020/1/4 2020/1/10 2020/1/16 2020/1/22 2020/1/28 2020/2/3 2020/2/9 2020/2/15 2020/2/21 2020/2/27 2020/3/4 2020/3/10 2020/3/16 2020/3/22 2020/3/28 2020/4/3 2020/4/9 2020/4/15 2020/4/21 2020/4/27 2020/5/3 2020/5/9 2020/5/15 2020/5/21 2020/5/27 2020/6/2 2020/6/8 2020/6/14 2020/6/20 2020/6/26 2020/7/2 2020/7/8 2020/7/14 2020/7/20 2020/7/26 2020/8/1 2020/8/7 2020/8/13 2020/8/19 2020/8/25 2020/8/31 2020/9/6 2020/9/12 2020/9/18 2020/9/24 2020/9/30 2020/10/6 2020/10/12 2020/10/18 2020/10/24 2020/10/30 2020/11/5 2020/11/11 2020/11/17 2020/11/23 2020/11/29 2020/12/5 2020/12/11 2020/12/17 2020/12/23 2020/12/29 | 61 |
| 2021 | 2021/1/4 2021/1/10 2021/1/16 2021/1/22 2021/1/28 2021/2/3 2021/2/9 2021/2/15 2021/2/21 2021/2/27 2021/3/5 2021/3/11 2021/3/17 2021/3/23 2021/3/29 | 15 |

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
