# Peer review of "Water Body Mapping Using Long Time Series Sentinel-1 SAR Data in Poyang Lake"

_water, doi:10.3390/w14121902_

Round 1

Reviewer 1 Report

In the present form, the paper requires major revisions:

(1) Why did authors used deep learning/CNN models compared other soft computing models? Introduction section needs these justifications

(2) Authors have studied the relationship between water body surface area and water level, and studying wetland environment changes using Sentinel-1 SAR data. Why did authors use Sentinel-1 SAR rather Sentinel-2?? This justification is needed.

(3) Literature review needs improvements by using Sentinel-1 and flood monitoring modeling: "Computers & Geosciences, 105045, 2022" and "Water 13 (21), 3115, 2021".

(4) Why did author consider study area and data?

Reviewer 2 Report

It is the interesting paper worthy publishing. Please quote additionally:

Journal: Remote Sens., 2022
Volume: 14
Number: 1056

Article: Sentinel-1 Satellite Radar Images: A New Source of Information for Study of River Channel Dynamics on the Lower Vistula River, Poland
Authors: by Klaudia Kryniecka, Artur Magnuszewski and Artur Radecki-Pawlik
Link: https://www.mdpi.com/2072-4292/14/5/1056

Reviewer 3 Report

Dear Authors,

your work is interesting and can help in better understanding how to use satellite information for detecting changes in (standing) water bodies. However, in my opinion, the present version is not ready for publication, and major changes should be integarted.

Below you can find very general comments, but please refer to the attached pdf for more specific hints.

Introduction: besides providing an overview of what SAR is, I suggest describing more in detail the study rationale (e.g., why the Poyang Lake? why SAR data?).

Secs. 2 and 3: I suggest combining these two sections, in a more common "Materials and Methods" section

Methodology: as you are applying some well-known methods, please acknowledge it by adding the proper references

Results and Discussion: I prefer seeing two separated sections, with the Discussion section focused on comparing the proposed approach and the results with similar studies available in the literature. This could be very important in stressing the research novelty.

You derived a water level/area relationship using multiple years. But you also said that sometimes sediments are also entering the lake in a large quantity. I assume that you imposed fixed bathymetry to derive the level/area relationship, and I am wondering if you can comment a bit more on the (possible) influence of bathymetric changes due to sediments.

Figures: please use always the same fonts

Round 2

Reviewer 1 Report

Although authors have made efforts to significantly revise the paper, the comparisons section still needs improvements: 

1- Using the 4th and 5th references, authors are strongly recommended to improve comparisons of the present results with literatures. All the comparisons should be prepared in details in terms of qualitative and quantitatively.

Reviewer 3 Report

Dear Author,

I would like to thank you very much for having addressed my comments. In my opinion, the present version is suitable for publication.

Author Response

This manuscript is a resubmission of an earlier submission. The following is a list of the peer review reports and author responses from that submission.

Round 1

Reviewer 1 Report

Comments:

This paper presents a methodology for obtaining water levels in a lake from Satellite Images. The method is compared with real data from a gauge station and good correlation is obtained.  

The paper needs some revision in English language but the method is well founded on scientific basis. Some extra work has to be done in Figures and in text. Apart from these minor changes, the paper needs some further explanation about how to correlate the area from Satellite images and the depth/height of water in the lake. Some figure showing this relation would be very helpful.

Abstract:

Page 1, line 1 is difficult to read. Please review the English.

Paper:

Figure 1a) what is this? Is this China? The river Catchment? Please add any geographical reference (seas, countries, etc).

Figure 2 would be better turned 90 degrees to have the words in the right direction. I do not see the point to have the figure in the current view. Also it is not clear the meaning of this Figure, it says statistics but it only shows numbers, not percentages.

Page 4, line 127 refers to Table A1 and this is in page 20 and in the middle of the reference list. Please locate Table A1 before or latter in an Appendix.

Figure 3. It is not clear the meaning of this Figure, it says statistics but it only shows numbers, not percentages. It good be better as “Number of observations per month...”

Figure 5 has to be improved with different colours for data, methods, calculations, etc.

Page 6, line 186 is difficult to understand, please review the English.

Page 7, line 191 “The workflow included 9 steps...” or “The workflow includes 9 steps”. Please, in general review the English, there are some typo errors and difficult sentences.

Figure 11. What is a), b) and c), the blue line is difficult to identify. What is each Picture. Why different colours. Please clarify the figure.

Page 12, paragraph from line 314-324. This is not clear, the water level is rising up and the images show the opposite? Please explain this cause it is difficult to understand.

Figures 12, 13 and 14. They have the same caption but no further explanation of the differences between figures. Please add any more information. What are the differences between the figures? What is the meaning of each figure? Only the dates, what do you want to show in these figures?

Reference List:

Some references include the year in bold letters, others do not. Please review this.

Reviewer 2 Report

The submitted research work investigate the use Sentinel-1 SAR data for mapping of Poyang Lake which is the largest freshwater lake in China. I found this paper very interesting where Several technical aspects were nicely implemented and explained sufficiently. Undoubtedly, authors invested huge amount of time and have made a great effort to produce this high-quality of research which is clearly structured and the language used is largely appropriate. As final decision, I see that this manuscript in its form and level deserves to be accepted for publication in MDPI-RS. 

Reviewer 3 Report

Unfortunately, I did not understand the reason for your study. Radar technology is applied but no justification why this method is needed and what the advantage is over other methods using the same satellite date. All you have written is very specific to Poyang, I do not see how the method can be applied to other water bodies, since your training set is the lake only. Moreover by declining to share any of your input data or your source code, no other scientist/operator will be able to work with your method or build on this. I strongly encourage  you to publish data/source codes. If data is confidential the information is worth only for your funding agency, not the community.

If you consider to make your work available, I provided some comments in the PDF.

Especially your figure 19 needs more interpretation. Usually with increasing water level also the area increases, except in gorges.

Round 2

Reviewer 3 Report

The revised version of the manuscript has very much improved since the first version and is now more easy to read. My main concern is the targeted audience of the paper. Over most parts it reads as a technical description and not an environmental study. But to fall in the technical paper, it is absolutely necessary to provide supplementary information, data and software. The paper describes the performance of the newly developed SARProcMod and WaterUNet software by comparing the results to a gauging station. Without providing the software and data, the outcome of the paper remains a technical exercise only.